# Temporal and Spatial Dynamics of EEG Features in Female College Students with Subclinical Depression

**DOI:** 10.3390/ijerph19031778

**Published:** 2022-02-04

**Authors:** Shanguang Zhao, Siew-Cheok Ng, Selina Khoo, Aiping Chi

**Affiliations:** 1Centre for Sport and Exercise Sciences, Universiti Malaya, Kuala Lumpur 50603, Malaysia; s2013074@siswa.um.edu.my (S.Z.); selina@um.edu.my (S.K.); 2Department of Biomedical Engineering, Faculty of Engineering, Universiti Malaya, Kuala Lumpur 50603, Malaysia; 3Institute of Physical Education, Shaanxi Normal University, Xi’an 710119, China

**Keywords:** depression, microstate, omega complexity, resting-state EEG, visual processing

## Abstract

Synchronization of the dynamic processes in structural networks connect the brain across a wide range of temporal and spatial scales, creating a dynamic and complex functional network. Microstate and omega complexity are two reference-free electroencephalography (EEG) measures that can represent the temporal and spatial complexities of EEG data. Few studies have focused on potential brain spatiotemporal dynamics in the early stages of depression to use as an early screening feature for depression. Thus, this study aimed to explore large-scale brain network dynamics of individuals both with and without subclinical depression, from the perspective of temporal and spatial dimensions and to input them as features into a machine learning framework for the automatic diagnosis of early-stage depression. To achieve this, spatio–temporal dynamics of rest-state EEG signals in female college students (*n* = 40) with and without (*n* = 38) subclinical depression were analyzed using EEG microstate and omega complexity analysis. Then, based on differential features of EEGs between the two groups, a support vector machine was utilized to compare performances of spatio–temporal features and single features in the classification of early depression. Microstate results showed that the occurrence rate of microstate class B was significantly higher in the group with subclinical depression when compared with the group without. Moreover, the duration and contribution of microstate class C in the subclinical group were both significantly lower than in the group without subclinical depression. Omega complexity results showed that the global omega complexity of β-2 and γ band was significantly lower for the subclinical depression group compared with the other group (*p* < 0.05). In addition, the anterior and posterior regional omega complexities were lower for the subclinical depression group compared to the comparison group in α-1, β-2 and γ bands. It was found that AUC of 81% for the differential indicators of EEG microstates and omega complexity was deemed better than a single index for predicting subclinical depression. Thus, since temporal and spatial complexity of EEG signals were manifestly altered in female college students with subclinical depression, it is possible that this characteristic could be adopted as an early auxiliary diagnostic indicator of depression.

## 1. Introduction

Depression is a common disorder, with its main symptoms being depressed mood, lack of interest in daily life, insomnia, and inability to enjoy life [1]. Numerous neuroimaging studies have revealed that depression appears to be a psychiatric disorder caused by abnormalities in brain function or structure [2,3,4]. It was found that synchronization of dynamic processes in structural networks connect the brain in a wide range of temporal and spatial scales, forming a dynamic and complex functional network [5]. A multiscale (spatial and temporal) understanding of large-scale brain network dynamics facilitates the elucidation of the underlying pathophysiology of depression.

Large-scale networks are dynamically re-organized on subsecond time scales to function efficiently [6]. An electroencephalogram (EEG) is a non-invasive tool with a high temporal resolution capable of detecting the spontaneous and rhythmic electrophysiological activity of cortical neuron populations [7]. Microstate and complexity are two reference-free EEG measurement methods. Microstate reflects the temporal dynamics of the functional brain network, while omega complexity reflects the spatial dynamics of the functional brain network [8]. EEG microstates are defined as transient (60–120 ms) quasi-steady electric potential scalp topography that provide coordinated and synchronized current activity of many simultaneously activated neurons [9]. The advantage of microstate analysis is its ability to characterize the spatial organization and temporal dynamics of large-scale cortical activity with high temporal resolution, taking into account the signals recorded in various regions of the cerebral cortex [10]. Numerous experiments have shown that the four resting-state microstates classes (Class A, B, C, and D) of the brain can be systematically extracted by cluster analysis methods, which can map approximately 80% of global variance [11].

Each microstate class represents a parallel information processing system in a distributed neural network. Microstate class A in the bilateral superior temporal gyrus and middle temporal gyrus are associated with negative activation of BOLD, which play a vital role in speech processing [12]. Microstate class B reflects visual resting-state networks and involves representational thinking [13]. Microstate class C, relating to the default mode network, is capable of integrating endoreceptive information with emotional salience to form subjective representations [14]. Microstate class D is associated with activities of the lateral dorsal and ventral regions of the frontal and parietal cortices, which are part of the dorsal attentional network [15,16]. Characteristics of microstates can be described in terms of duration, occurrence rate, contribution, and transition probabilities [17]. The duration of a microstate is measured in the time difference between the start of the first microstate class and the start of the next microstate class, reflecting the stability of underlying neural components. The rate of occurrence of microstates indicates the number of occurrences of a class of microstate per second, reflecting the tendency of the underlying neural generator to be activated. Contribution is the average percentage of time each microstate covers, calculated from its duration and occurrence. Transition probability between microstates is the probability of transferring from one microstate to another and is usually interpreted as the asymptotic behavior of transitions between microstates [18]. Previous studies have found that the duration of microstate class C is shorter in patients with depression, while the occurrence rate of microstate class B is higher [19,20]. A shorter duration of a microstate indicates that information processing of some cognitive processes may be terminated in advance. The higher the occurrence rate, the more frequently the basic steps of the cognitive process must be repeated to complete certain mental processes [21].

There are a variety of EEG measurements based on chaos or nonlinear systems theory that characterizes the physiological complexity of the human brain. These include approximate entropy, Taken’s estimator, Lyapunov exponent, and multiscale entropy [22,23]. It has been shown that intrinsic complexity enables adaptation of biological systems, while loss of complexity is strongly associated with cognitive impairment in patients with schizophrenia, depression, and Alzheimer’s disease [24]. However, the above methods can only describe the complexity of the neurophysiological time series of a single channel, but they cannot evaluate spatial complexity within the whole or regional brain.

Wackermann et al. (1993) proposed using the omega complex by combining principal component analysis and Shannon entropy calculations to quantify the global synchronization of frequency bands between spatially distributed brain regions. The omega complexity used in EEG signal analysis enables it to describe the spatial dynamics of the brain electrical field in the macro state by finding the correlation between the multiple channels and representing the final results as a single number. Omega complexity evaluates the degree of synchronization between spatially distributed processes by defining the entropy of λ-spectra of cross-spectral matrices of the EEG data for each frequency point via Fourier transform. In addition, omega complexity is calculated for the kinetic characteristics of multichannel signals with two or more channels, which can be flexibly adapted to analyze regional or overall changes, reflecting the signal’s spatial synchronization. Research has shown that omega complexity was sensitive to cognitive processes in different types of mental illness. For example, some studies found a significant increase in anterior omega complexity in patients with schizophrenia, suggesting loosened cooperation or coordination of brain processes in the forebrain regions active in schizophrenia [25,26]. Another study of male adolescents with mild spastic diplegia also found increased omega spatial complexity of EEG signals [27]. Omega complexity was lower in a study of patients with epilepsy [20,22]. Although altered spatial complexity has been found in neurological disorders, it has not been studied in individuals with depression. Several studies have suggested abnormalities in large-scale brain networks of patients with depression, including inadequate connectivity in the frontoparietal network [4], reward circuits centered in the ventral striatum [28], and reduced functional connectivity between the frontoparietal and cingulo-opercular networks [29]. Complexity is a characteristic of a healthy biological system, meaning that other psychopathologies are accompanied by lower complexity, and functional connectivity is altered in depression. Accordingly, we hypothesize that brain functional network complexity is reduced in college students with subclinical depression (ScD). We also hypothesized that the altered spatio–temporal dynamics are related to the degree of depression among female college students.

Although the combined use of EEG microstates and omega complexity to study mental disorders has only been conducted in one study, to our knowledge [27], it has proven valuable in investigating pathophysiology profiles associated with neurological or psychiatric disorders. ScD, considered prodromal/premorbid to major depressive disorder (MDD), is a description applied when an individual experiences depressive symptom that are not severe or persistent enough to be diagnosed as MDD [30,31]. ScD has become widespread among college students worldwide [32,33,34]. It was also found that individuals with ScD had a five-fold increased risk of experiencing their first episode of MDD [35]. In addition, females were more affected by stress and anxiety than males, often leading to a higher prevalence of ScD [36,37]. In consequence, this study focuses on female college students with ScD and intends to provide a basis for preventing the development of DMM.

Machine Learning (ML) is the core of artificial intelligence. The purpose of ML is to enable computers to learn from existing data and information, obtaining potential patterns in the data that could be applied to the analysis and prediction of unknown data [38,39]. With powerful data processing and excavation capability, ML methods provide advantages for predicting various diseases [40,41]. ML algorithms applied to depression are random forests, K-nearest neighbour, Bayesian networks, and support vector machine (SVM) classifiers [42,43]. SVM can find the best compromise between model complexity and learning ability based on limited sample information, minimizing structural risks to obtain the best learning generalization ability [44]. Moreover, SVM is better able to deal with small-sample, nonlinear, and high-dimensional issues [40]. Since the sample size of this study is relatively small, the SVM model was used for binary classification. According to the class labels, the SVM model can classify the feature space based on a ‘hyperplane’ that distinguishes students with and without ScD.

To explore the dynamics of large-scale brain networks in ScD at multiple scales (spatial and temporal), this study aimed to distinguish differences in features between individuals with and without ScD using EEG microstates and omega complexity analysis. Based on these different features, the secondary aim was to perform dichotomies by ML to verify that classification performance was better for combining spatial and temporal information than single-dimensional information.

## 2. Methods

### 2.1. Participants

Study participants were first-year female students at Shaanxi Normal University. They were assessed by two psychologists from the University Counseling Center using the Self-Rating Depression Scale (SDS) [45] and the Beck Depression Scale-II (BDI-II) [46]. Chinese BDI-II is a 21-item self-reporting inventory to assess depressive symptoms, having good reliability (α = 0.911) and validity in college students [47]. Inclusion criteria for the ScD group were BDI-II scores between 14 and 27, and having sufficient visual and auditory levels to complete the required experiments [48]. Inclusion criteria for the healthy controls group (HCs) were BDI-II scores below 13 and SDS scores below 15. Exclusion criteria for both groups included: (1) a history of traumatic brain injury or depression; (2) suicide attempt; (3) taking psychiatric medications (including antidepressants, mood stabilizers, antipsychotics, and benzodiazepines); (4) having physical comorbidities (e.g., cerebrovascular disease and cancer), and (5) having mental comorbidities (e.g., schizophrenia, bipolar disorder, and post-traumatic stress disorder). We performed a priori power analysis using G*Power 3.1. The sample size of 36 for each group was sufficient. A priori power analysis for mixed repeated measure ANOVA suggested obtaining a large effect with 0.90 power (1 − β) and 0.05 Type I error rate, and the number of repetitions was 4. A total of 40 female students with ScD (mean age = 18.72 years, SD = 0.36 years) and 38 healthy students of similar age were recruited. According to the Edinburgh Handedness Inventory, all participants were right-handed [49]. Demographic data of participants are shown in Table 1. There were no statistical differences for age, BMI, and education level between the two groups.

The study was conducted in accordance with the Declaration of Helsinki principles and all procedures were carried out with adequate understanding and written informed participant consent. Research ethics approval was obtained from the Ethics Committee of Shaanxi Normal University (202116010). All participants gave written informed consent and received financial compensation for participating in the study.

### 2.2. EEG Acquisition and Preprocessing

Participants were seated in a comfortable experimental laboratory and exposed to limited sound and appropriate lighting. During the experiment preparation phase, participants were asked to remain relaxed and avoid strenuous movements of the head and hands throughout the entire experiment. In total, 4 min of resting-state EEG data were collected with eyes closed and eyes opened. EEG data were recorded with 32 electrodes following the 10/10 international electrode placement system (Neuroscan Inc., Abbotsford, VIC, Australia). The EEG acquisition system applied a 0.1–100 Hz bandpass filter to the signal and digitized it with a sampling rate of 1024 Hz. Electrodes CPz and AFz were used as reference and ground. Vertical electrooculogram (EOG) was recorded with electrodes placed above and below the left eye, and the horizontal EOG was recorded with electrodes placed lateral side of both eyes. All electrode impedances were maintained below 10 kΩ.

The study analyzed EEG data in the closed-eye state as there was excessive eye-movement artifacts in the open-eye state. Raw EEG data was pre-processed using EEGLAB (Version R2013b, San Diego, CA, USA), an open-source toolbox running on MATLAB environment (Version R201 3b, MathWorks, Natick, MA, USA). A 0.5–45 Hz bandpass filter, as well as a 50 Hz notch filter, were applied to the EEG data using a finite impulse response filter. Thereafter, the EEG waveforms were epoched into segments of 2-s duration and remontaged to an average reference. An independent component analysis (ICA) procedure was used to identify and extract artifact components, and remove the segment of the sources containing eye blink artifacts, eye movement, and EMG artifacts (High-frequency signals). EEG data processing and analysis process is shown in Figure 1.

### 2.3. EEG Microstate Analysis

In microstate analysis, the multichannel EEG signal is considered a series of instantaneous topographies of electric potentials. The peak position of the global field power (GFP) curve represents the moment of the strongest field intensity and the highest topographic signal-to-noise ratio. The GFP is equivalent to the root mean squared of electrodes at that time point:$$GFT(t)=\sqrt{\frac{\sum_{i=1}^{n}(v_{i}(t)-\bar{v}(t){)}^{2}}{n}}$$
where *v_i_*(*t*) is the voltage at electrode *i* at time *t*, $\bar{v}\left(t\right)$ is the mean voltage across all electrodes at time *t*, and *n* is the number of electrodes.

To begin, the processed EEG data was digitally bandpass filtered at 2 to 20 Hz (Figure 1) [17,50]. Then, the GFP of each participant across 30 electrodes was calculated, and the multichannel EEG topography was obtained based on the GFP. Lastly, the EEG topography of participants was submitted to the Topographic Atomize & Agglomerate Hierarchical Clustering algorithm, which identified clusters with similar topology. The number of clusters was predetermined to encompass four classes (A/B/C/D). Polarity of EEG topography was ignored. Four parameters—duration, occurrence, contribution, and transition—were calculated for each microstate. Global interpreted variance (GEV) index was used to judge the fitting effect. GEV is calculated by calculating the similarity between EEG sample points and their prototype microstates. A higher GEV value indicates a better fit.

### 2.4. Omega Complexity Analysis

The 30-channel EEG data from each subject were transformed to the frequency domain using Fourier Transform. The omega complexity of the seven frequency bands was calculated as the average value within each frequency limit: delta (1–3.5 Hz); theta (4–7.5 Hz); alpha-1 (8–10 Hz); alpha-2 (11–13 Hz); beta-1 (13–20 Hz); beta-2 (20–30 Hz); and gamma (30–45 Hz). Studies have shown that the complexity of anterior and posterior regions have different functional significance [25]. Thus, the global complexity and regional complexity of each frequency band were calculated. Global complexity was obtained by calculating the cross-spectral matrix of the 30 × 30 of all electrodes. Regional complexity was obtained by calculating a 7 × 7 cross-spectral matrix using electrodes in the prefrontal (i.e., electrode Fp1, Fp2, F7, F3, Fz, F4 and F8) and posterior (i.e., electrode T5, T6, P3, P4, Pz, O1, and O2) regions. Principal component analysis of these cross-spectral matrices yielded a spectrum of eigenvalues. These were normalized to assess the relative contribution of each principal component to the total variance. The calculation process of omega was as follows. First, the cross-spectral matrix *C* of global complexity and local complexity was calculated, respectively:$$C=\frac{1}{N}\sum_{i=1}^{N}u_{i}*u_{i}^{T}$$
where *K* is the number of electrodes, *N* is the EEG signal length.

Then the eigenvalue *λ*_1_, …, *λ_k_* of the matrix *C* was calculated. Next, the normalized feature $\lambda_{i}^{\prime }$ was calculated.
$$\lambda_{i}^{\prime }=\frac{\lambda_{i}}{\sum \lambda_{i}}$$

According to the definition of omega:$$log\left(\Omega \right)=-\sum \lambda_{i}^{\prime }*log\left(\lambda_{i}^{\prime }\right)$$

Therefore, the omega spatial complexity can be calculated:$$Omega=exp\left\{-\sum_{i=1}^{K}\lambda_{i}^{+}\log\left(\lambda_{i}^{\prime }\right)\right\}$$

The omega complexity can be viewed as a measure of the spatial complexity of a given set of EEGs, which ranges in value from 1 to *K*. A smaller omega value indicates that the number of modes existing between the computed signals is small, with a single-mode, and a high degree of synchronization; a larger omega value indicates that the number of modes existing between the computed signals is large and the degree of synchronization is poor. For example, if the value of omega tends to *K*, it is known from the knowledge of information theory that the calculated signal has a uniform distribution of modes with the highest number of modes, the highest complexity, and the worst synchronization; on the contrary, if the value of omega tends towards 1, it indicates that the distribution of the signal has only one mode, and the highest synchronization is achieved. The omega complexity values for each frequency point were averaged over the epochs (Gao et al., 2017; Kondakor et al., 2005).

### 2.5. Support Vector Machine

All analyses were performed in R. Before using SVM classification, all features were converted into sequences with mean 0 and 1 by the normalization method. In this study, the linear kernel and the Gaussian radial basis kernel functions were selected as kernel functions, and the hyperparameters were tuned by grid search. Nested cross-validation was used to search for the super parameters by estimating the generalization error of the underlying model to obtain the best model parameters. The classification performance was evaluated based on nested cross-validation (remaining one cross-validation for the outer layer, five-fold for the inner layer). The inner layer was five-fold cross-validation, splitting the data into five parts: four for training and one for testing (80% for training and 20% for testing), leaving sufficient test samples to ensure that useful information about accuracy could be provided. In addition, the receiver operating characteristic (ROC) curve was performed for evaluating SVM models. The area under the curve (AUC) of the ROC curve was 0.5 for random probability prediction and 1 for perfect prediction.

### 2.6. Statistical Analysis

Statistical analysis was performed using SPSS (23.0; SPSS, Inc., Chicago, IL, United States). The Wilcoxon test was used for non-normally distributed data. Two-way analyses of mixed repeated-measures variance (ANOVA) with a between-subject factor for the group (ScD and HCs) and a within-subject factor for microstate classes (A/B/C/D) were performed for the microstate duration, occurrence, and contribution. Two-way analyses of mixed repeated-measures ANOVA with between-subject factor for the group (ScD and HCs) and within-subject factor for microstate transition probability 12 (A -> B, A -> C, A -> D, B ->A, B-> C, B -> D, C ->A, C -> B, C->D, D ->A, D -> B, D -> C) were performed for the microstate transition probability. Two-way ANOVA with between-subject factor for the group and within-subject factor for scalp region (anterior, posterior) was also performed for the omega complexities in each frequency band. A Greenhouse–Geisser correction of the ANOVA assumption of sphericity was applied where appropriate. The Bonferroni-correction method was used to correct multiple comparisons. In addition, an independent sample t-test was performed for the global omega complexity. The level of significance was set at *p* < 0.05. Effect size in all ANOVA analyses was reported by partial eta squared (*η*^2^), where 0.05 represents a small effect, 0.10 represents a medium effect, and 0.20 represents a large effect [51]. Cohen’s *d* was computed to estimate the effect size of post hoc tests.

## 3. Results

### 3.1. EEG Microstate

In this study, the four microstate classes accounted for a mean of 81.38% (SD = 3.58%) of the data variance across the ScD group and 82.12% (SD = 4.02%) of the data variance across the HCs group.

Microstate occurrence. The main effect of this microstate class was significant, *F* (3, 183) = 4.40, *p* = 0.008, *η*^2^ = 0.067. Post hoc *t*-tests revealed a significantly higher frequency of microstate class A than microstate class D. The main effect of the group was not significant, *F* (1, 61) = 1.84, *p* = 0.184, *η*^2^ = 0.029. The interaction effect between group and microstate class interaction was significant, *F* (3, 183) = 2.87, *p* = 0.038, *η*^2^ = 0.035. Simple effects analysis found a significant difference between the two groups of students in microstate class B, *p* < 0.05. Further post hoc analysis found that microstate class B had a higher frequency for the ScD than the HCs group.

Microstate duration. The main effect of this microstate class was significant, *F* (3, 183) = 1.71, *p* = 0.168, *η*^2^ = 0.027. The main effect of the group was not significant, *F* (1, 61) = 2.17, *p* = 0.146, *η*^2^ = 0.034. The interaction effect between group and microstate class was significant, *F* (3, 183) = 1.56, *p* = 0.202, *η*^2^ = 0.025. Post hoc tests revealed that the durations of microstate class C were significantly longer for the HCs group compared to the ScD group (*t* = 2.891, *df* = 61, *p* = 0.005, Cohen’s *d* = 0.74).

Microstate contribution. The main effect of this microstate class was significant, *F* (3, 183) = 3.16, *p* = 0.026, *η*^2^ = 0.049. The main effect of the group was not significant, *F* (1, 61) = 1.85, *p* = 0.179, *η*^2^ = 0.029. The interaction effect between group and microstate class was not significant, *F* (3, 183) = 2.11, *p* = 0.100, *η*^2^ = 0.033. Follow-up *t* tests indicated significant differences in the contribution of microstate class C for the HCs group compared to the ScD group (*t* = −2.76, *df* = 61, *p* = 0.008, Cohen’s *d* = 0.706). The results are shown in Table 2.

Microstate transitions probability. The main effect of microstate transition pairs was significant, *F* (11, 371) = 3.69, *p* = 0.026, *η*^2^ = 0.049. The main effect of the group was not significant, *F* (1, 61) = 0.360, *p* = 0.55, *η*^2^ = 0.006. The interaction effect between group and microstate transition probability was significant, *F* (11, 371) = 2.50, *p* = 0.004, *η*^2^ = 0.039. The results of post hoc tests indicated the microstate transition pairs significantly increased from microstate class A to microstate class B (*t* = 3.288, *df* = 61, *p* = 0.002, Cohen’s *d* = 0.827) and from microstate class B to microstate class A (*t* = 2.879, *df* = 61, *p* = 0.005, Cohen’s *d* = 0.737) for the ScD group than for the HCs group. The results also significantly increased from microstate class C to microstate class D (*t* = −2.399, *df* = 61, *p* = 0.019, Cohen’s *d* = 0.642) and class D to class C (*t* = −2.238, *df* = 61, *p* = 0.029, Cohen’s *d* = 0.573) for the HCs group than for the ScD group. The other microstate transition probabilities were not significant between the ScD and HCs (A→C: *p* = 0.20; A→D: *p* = 0.54; B→C: *p* = 0.44; B→D: *p* = 0.27; C→A: *p* = 0.25; C→B: *p* = 0.50; D →A: *p* = 0.71; D→B: *p* = 0.22). Significant results for microstate transition probability are shown in Figure 2.

### 3.2. Omega Complexity

We first compared the global omega complexity of the two groups using an independent samples *t*-test, which is shown in Table 3. The results revealed that global omega complexity has a significant reduction in the beta-2 (*t* = −2.452, *df* = 61, *p* = 0.017\0.05, Cohen’s *d* = 0.63) and gamma bands (*t* = −2.546, *df* = 61, *p* = 0.013\0.05, Cohen’s *d* = 0.65) for the ScD group compared to the HCs group.

Then, a 2 (ScD vs. HCs) × 2 (scalp region: anterior and posterior) mixed ANOVA was conducted for regional omega complexities at seven EEG frequency bands (i.e., delta, theta, alpha-1, alpha-2, beta-1, beta-2, and gamma), which is shown in Figure 3.

There was no significant main effect and interaction effect in δ band.

θ band: results showed that the main effect of the group was significant, *F* (1, 61) = 31.141, *p* < 0.000, *η*^2^ = 0.338; post hoc analysis revealed that the regional omega complexities were significantly lower for the ScD group (*M* = 2.68, *SE* = 0.08) compared to the HCs group (*M* = 3.00, *SE* = 0.08). The main effect of the scalp region was also significant, *F* (1, 61) = 3.69, *p* = 0.026, *η*^2^ = 0.049. The posterior regional (*M* = 2.62, *SE* = 0.71) omega complexities were significantly lower compared to the anterior regional (*M* = 3.06, *SE* = 0.07). There was no significant interaction effect, *F* (1, 61) = 2.56, *p* = 0.115, *η*^2^ = 0.040.

α-1 band: results showed that the main effect of scalp region was significant, *F* (1, 61) = 132.52, *p* < 0.000, *η*^2^ = 0.685; there was no significant main effect of group, *F* (1, 61) = 0.832, *p* = 0.365, *η*^2^ = 0.013. There was a significant interaction effect between group and scalp region *F* (1, 61) = 5.00, *p* = 0.029, *η*^2^ = 0.076; post hoc analysis revealed that omega complexities were significantly lower in the posterior region (*M* = 1.71, *SE* = 0.54) compared to the anterior region (*M* = 2.50, *SE* = 0.07).

α-2 band: results showed that the main effect of scalp region was significant, *F* (1, 61) = 201.16, *p* < 0.000, *η*^2^ = 0.767; post hoc analysis revealed that omega complexities were significantly higher in the posterior region (*M* = 2.40, *SE* = 0.06) compared to the anterior region (*M* = 1.57, *SE* = 0.04). There was also a significant interaction effect between group and scalp region *F* (1, 61) = 7.87, *p* = 0.007, *η*^2^ = 0.114. Further comparative analysis found that omega complexities were significantly higher for the HCs group (*M* = 2.51, *SE* = 0.45) compared to the ScD group (*M* = 2.29, *SE* = 0.45), *p* = 0.057.

β-1: results showed that the main effect of scalp region was significant, *F* (1, 61) = 101.06, *p* < 0.000, *η*^2^ = 0.624; post hoc analysis revealed that omega complexities were significantly higher in the posterior region (*M* = 2.88, *SE* = 0.05) compared to the anterior region (*M* = 2.21, *SE* = 0.06), *p* < 0.000. There was no significant main effect of group and interaction effect between group and scalp region.

β-2: results showed that the main effect of scalp region was significant, *F* (1, 61) = 11.04, *p* = 0.002, *η*^2^ = 0.153; post hoc analysis revealed that omega complexities were significantly higher for the posterior region (*M* = 3.14, *SE* = 0.06) compared to the anterior region (*M* = 2.88, *SE* = 0.08). There was also a significant main effect of group, *F* (1, 61) = 4.63, *p* = 0.035, *η*^2^ = 0.071. Further comparative analysis found that omega complexities were significantly higher for the HCs group (*M* = 3.30, *SE* = 0.40) compared to the ScD group (*M* = 2.98, *SE* = 0.52) in the posterior region, *p* = 0.01.

γ band: results showed that the main effect of scalp region was significant, *F* (1, 61) = 5.55, *p* = 0.022, *η*^2^ = 0.083; post hoc analysis revealed that omega complexities were significantly higher in the anterior region (*M* = 3.38, *SE* = 0.09) compared to the posterior region (*M* = 3.19, *SE* = 0.07), *p* = 0.022. There was also a significant main effect of group, *F* (1, 61) = 5.93, *p* = 0.018, *η*^2^ = 0.089. Further analysis found that omega complexities were significantly higher in the HCs group (*M* = 3.46, *SE* = 0.10) compared to the ScD group (*M* = 3.11, *SE* = 0.10) in the anterior region, *p* = 0.018. There was no significant interaction effect between group and scalp region *F* (1, 61) = 1.150, *p* = 0.288, *η*^2^ = 0.019.

### 3.3. Correlations Results

Results of Spearman’s rank correlation revealed a significant negative correlation between the duration (*r* = −0.45; *p* = 0.007) and contribution (*r* = −0.39; *p* = 0.024) of the microstate class A and the BDI-II score (*r* = −0.45; *p* = 0.007). A significant positive association between the microstate transition (B→D) and the BDI-II score (*r* = 0.38; *p* = 0.026) scores was also revealed. Furthermore, a significant positive association between global omega complexity in θ (*r* = 0.36; *p* = 0.034) and γ (*r* = 0.39; *p* = 0.023) bands and the BDI-II score scores was found. Finally, the results of Spearman’s rank correlation revealed a significant positive association between anterior regional omega complexity in θ (*r* = 0.35; *p* = 0.045), β-1 (*r* = 0.40; *p* = 0.02) and γ (*r* = 0.35; *p* = 0.044) bands and the BDI-II score scores.

### 3.4. SVM Results

Microstate indicators (the occurrence of microstate class B and the duration and contribution of microstate class C) and spatial complexity (the regional complexity of θ, β2 and γ bands and the global complexity of β2 and γ) were taken for ML analysis. Using the microstate parameter, the accuracy of different kernel functions in the SVM model were found to be 62% (gaussian) and 63% (linear), respectively. The classification accuracy of the SVM for omega parameters was 60%. The accuracy of the combination of the two indices were 75% and 76%, respectively, significantly higher than the accuracy rate of the single index, increasing by about 15%. The training set corresponding to the test set of the SVM was used to plot the ROC curve (Figure 4). Similarly, single indicator features were found to be less effective in prediction models of the ScD group; however, two indicators (AUC = 0.81 and 0.80) were more effective in prediction models of female college students with ScD. The classification performance of the SVM classifier based on the two kernel functions for each EEG indicator and the combination of the indicators are presented in Table 4.

## 4. Discussion

In this study, we investigated the spatio–temporal complexity of brain networks in female college students with and without ScD using EEG microstate indicators and omega complexity. We found a significant decrease in the occurrences of microstate B and a significant increase in the duration and contribution of microstate class C among students with ScD compared to those without. The change of spatial complexity was mainly manifested by a significant decrease in global omega complexity in the beta-2 and gamma bands, a significantly reduced prefrontal region in alpha-2 and gamma bands, and the posterior region of the beta-2 band among students with ScD, compared to those without. The SVM results showed that the differential index of the two analysis methods could distinguish female students with ScD from those without, with an AUC of 0.81 for the test sets. These results may contribute to the development of future auxiliary diagnostic methods.

### 4.1. EEG Microstates

Four microstates were separated from the two groups of EEG microstate sequences. The topological modes of the four microstates were generally consistent with previous studies [20,52]. The duration, rate of occurrence, and contribution of the four EEG-based microstates were calculated to assess the differences between groups. These properties of microstates have been frequently used in previous studies [53,54,55,56]. Microstates may correspond to specific categories of mental states, and these ongoing mental processes are known to affect the processing and response mode of incoming information [57].

We found that the occurrence rate of microstate class B was more frequent in the ScD than in the HCs groups, with no significant differences in duration or contribution. The source of microstate class B is located mainly in the occipital cortex and is associated with negative BOLD activation in the bilateral striate visual cortex, which is significant for visual processing [58]. Dedovic et al. (2005) explored the effects of psychological stress on individuals’ physiological and brain activation levels and found that the motor and visual cortices are activated when individuals are stressed Dedovic, et al. [59]. This may be indicative of the increased occurrence rate of microstate class B due to increased activation of bilateral striatal visual areas as a result of psychological stress.

We also demonstrated shorter durations and contribution of self-representation related microstate class C in the ScD group compared to the HCs group. Earlier studies have shown that the shortening of the duration of microstate class C is indicative of the cognitive decline in patients with MDD and has also been associated with neuropsychiatric symptoms [60,61]. Furthermore, microstate class C was reduced when new perceptions were constructed Müller, et al. [62]. Thus, a decrease in microstate class C was associated with difficulties in reconsidering previous cognitive-emotional assessments in response to certain environmental stimuli in female college students with depression.

We found a negative correlation between depressive symptom scores and the duration and contribution of microstate class A. However, the findings of Damborská et al. (2019) showed that the severity of depressive symptoms was positively correlated with the occurrence rate of microstate class A [19]. Murphy et al. (2020) also found that the occurrence rate of microstate class A increased for patients with MDD [20]. They also noted that the duration and contribution of microstate class D were negatively correlated with the severity of symptoms. No difference in microstate D was found between the two groups in this study. One study reported significantly lower duration, coverage, and contribution of microstate class D in participants with depression compared to healthy controls [20]. Consistent with previous findings, microstates classes A and B preferentially transfer between each other [63]. Microstates A and B are associated with lower-order sensory networks, whereas microstates C and D are associated with higher-order cognitive networks. One possible reason for this discrepancy is that our participants only had subclinical depression.

### 4.2. Omega Complexity

The human brain is complex, with multiple levels of temporal and spatial scales and consisting of interconnected feedback loops. Thus, studying individual brain signaling complexity may provide insight into understanding physiological complexity. Compared to traditional EEG analysis methods (e.g., spectral power, coherence, and nonlinear analysis), this study quantifies spatial complexity of EEG data by omega complexity, which assesses the number of independent electrophysiological sources and the overall degree of synchronization between spatially distributed brain regions. It also reveals a significant difference in EEG spatial complexity between ScD and HCs groups since the omega complexity in the ScD group differs globally and regionally from that of the HCs group. The anterior frontal region of theta band, the global and the posterior region of beta-2 band, and the global and anterior region of gamma band were significantly lower in the ScD group, as shown in Table 2.

Theta rhythms in the central frontal area are correlated with thought activity and concentration levels [64]. The beta rhythms are mostly found in the frontal lobe and are associated with cortical excitability, reflecting emotional and cognitive processes. One study found increased theta and beta rhythms synchrony in visual cognition in patients with depression and hypothesized that this was related to their attentional deficits [65]. In this study, we found that the theta band omega complexity of the anterior region and the global and posterior frontal omega complexity of the beta-2 bands in the ScD group was significantly lower than that of the HCs group, indicating a decrease in spatial complexity and an increase in synchronization. The present study further demonstrates that the attention deficit may be related to reduced spatial complexity. In addition, it has been shown that theta asymmetry in frontal–central brain regions in depressed patients may be related to their depressive and anxiety symptoms. In the present study, the complexity of global and anterior regions was positively correlated with depressive symptoms. The beta-1 band of the prefrontal region omega was positively correlated with the omega complexity depressive mood scale.

Gamma rhythms mainly appeared in the frontal and anterior central regions related to attention, excitement, perceptual processing, and condition perception. Li et al. (2015) found that during emotional processing, the brain network parameters of patients with depression showed a tendency to randomize [66]. The global complexity of the gamma band was significantly lower in this study, further demonstrating that the gamma network randomization trend of the group with depression has low spatial complexity. Gamma band global and anterior complexity is positively correlated with BDI scores, suggesting prefrontal complexity is associated with depressive mood regulation.

Previous studies have shown that higher spatial complexity is associated with higher information processing speed [67]. Low spatial complexity indicates that some of these functional brain regions are not activated, likely hindering information processing in the task. Furthermore, because the human nervous system works economically and efficiently, trials with higher spatial complexity may have relatively low levels of neural activation. Reduced global omega complexity demonstrates that the brains of female college students with subclinical depression are less coordinated during information transmission.

### 4.3. The Classification Results

Results of SVN based on different kernel functions were basically the same, but Gaussian functions improved efficiency by 1% over the results of linear kernel functions. The ML results showed that the predictive classification accuracy of female college students with subclinical depression was 81%. Mumtaz et al. (2018) used the decision tree method to classify the EEG frequency band power, and the accuracy reached 80% [68]. Earlier, Mumtaz et al. (2017) collected the EEG data of MDD in the resting state, showing that the classification accuracy of features extracted by wavelet transform was the highest, reaching 87.5% [69]. These studies have mainly focused on the diagnosis of MDD, whereas early recognition of subclinical depression may be particularly vital in preventing MDD.

### 4.4. Limitations

Although we found that the combination of microstates and omega was more accurate than a single indicator, a higher classification effect was not achieved due to the small sample. Moreover, the study did not adequately consider and control for confounding variables. Depression is a complex disorder with a complex etiology influenced by biological and social factors. In future studies, covariates such as parental education level and educational level correlated with subclinical depression, amongst others, should be included with strictly matched experimental and control groups.

## 5. Conclusions

Depression rates are increasing worldwide. Without early screening methods for depression, patients are only diagnosed when they develop MDD. In this study, EEG microstates and omega complexity analysis were used to obtain spatio–temporal dynamics of large-scale brain functional networks as features in female college students with and without ScD. The features of two groups with significant differences were then input into the SVM model for binary classification. The microstate results suggest that female college students with ScD may be due to psychological stress leading to an increased activation time of microstate class B-related visual cortex, while difficulties in the cognitive–emotional assessment are associated with less duration and contribution of microstate class C. Decreased Omega complexity in the theta, beta-2, and gamma bands indicate reduced functional connectivity within brain regions and may be associated with attention deficits. SVM results showed that the classification performance of spatio–temporal information features of EEG signals outperformed single-dimensional information features. The spatio–temporal complexity of the EEG signal is altered in the early stages of depression, which may provide a reference for future early ancillary diagnosis of depression.

## Figures and Tables

**Figure 1 ijerph-19-01778-f001:**
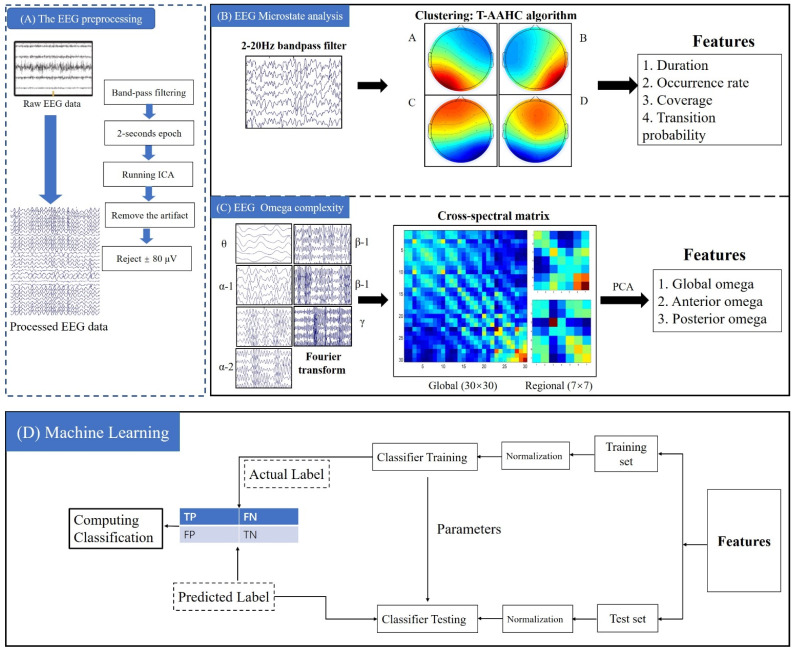
Analysis procedures for resting-state data sets. (**A**) Resting-state data pre-processing; (**B**) microstates analysis; (**C**) omega complexity analysis; (**D**) subclinical depression prediction.

**Figure 2 ijerph-19-01778-f002:**
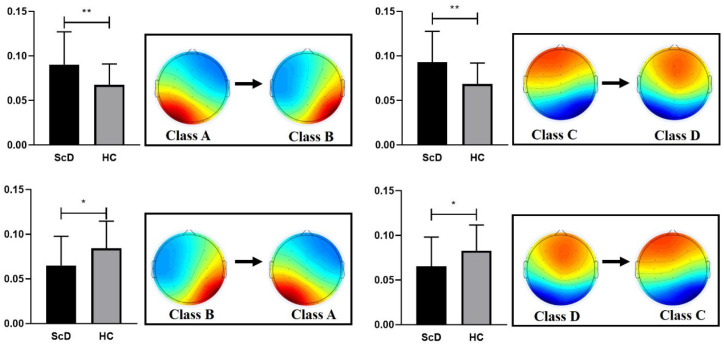
The four microstate classes significant transitions. *: *p* < 0.05, **: *p* < 0.01.

**Figure 3 ijerph-19-01778-f003:**
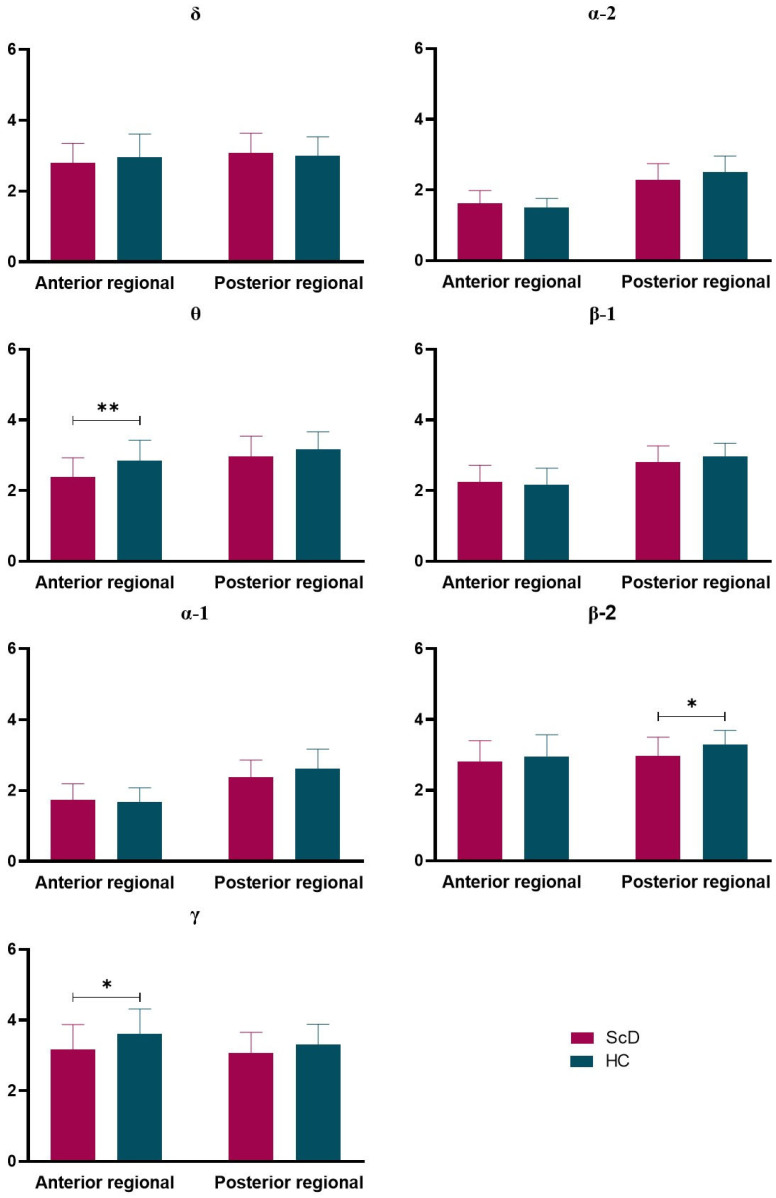
The mean regional omega complexities of seven EEG frequency bands (i.e., delta, theta, alpha-1, alpha-2, beta-1, beta-2, gamma). *: *p* < 0.05, **: *p* < 0.01.

**Figure 4 ijerph-19-01778-f004:**
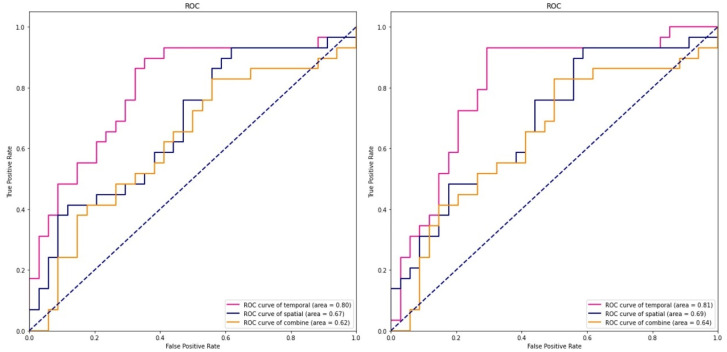
Receiver operating characteristic curves for spatial (microstate), temporal (omega complexity), and their combination using the SVM model based on Gaussian kernel function (left) and linear kernel function (right).

**Table 1 ijerph-19-01778-t001:** Demographic information of participants (Mean ± SD).

Variable	HCs (*n* = 38)	ScD (*n* = 40)
Age, years	18.72 ± 0.36	18.51 ± 0.42
Height, cm	162.71± 6.62	160.70 ± 6.73
Weight, kg	52.37 ± 4.72	50.00 ± 1.92
BMI, kg/m^2^	20.79 ± 2.73	19.43 ±1.61
SDS	10.57 ± 5.47	66.71 ± 5.38 ***
BDI-II	3.46 ± 0.73	24.86 ± 2.02 ***

Note: HCs, healthy controls; ScD, Subclinical depression; BMI, body mass index; SDS, Self-rating Depression Scale; BDI-II, Beck Depression Inventory II; ***: *p* < 0.001.

**Table 2 ijerph-19-01778-t002:** Microstate parameters of the ScD and HCs groups.

	Class A		Class B		Class C		Class D	
	Mean	SD	Mean	SD	Mean	SD	Mean	SD
Duration (ms)								
ScD group	71.50	12.13	64.53	8.99	62.91	12.52	67.77	34.68
HCs group	73.32	14.36	66.92	12.06	73.77	17.49	64.85	16.09
*t* (*df* = 61)	−0.546		−0.899		−2.891		0.415	
*p*	0.308		0.206		**0.005**		0.679	
Cohen’s *d*	0.14		0.22		0.74		0.11	
Occurrence/s								
ScD group	4.06	0.80	3.81	0.93	3.62	1.02	3.34	1.26
HCs group	3.67	0.83	3.27	0.98	4.03	0.74	3.28	0.92
*t* (*df* = 61)	1.882		2.256		−1.784		0.221	
*p*	0.065		**0.028**		0.078		0.826	
Cohen’s *d*	0.52		0.56		0.46		0.05	
Contribution (%)								
ScD group	28.98	8.79	24.62	7.54	23.14	9.12	23.25	14.06
HCs group	26.90	9.17	21.96	8.97	29.26	8.34	21.88	9.75
*t* (*df* = 61)	0.917		1.283		−2.758		0.442	
*p*	0.363		0.204		**0.008**		0.660	
Cohen’s *d*	0.23		0.32		0.70		0.11	

Bold values indicate statistical significance (*p* < 0.05).

**Table 3 ijerph-19-01778-t003:** Global omega complexity of the ScD and HCs groups.

	ScD	HCs	*t* (*df* = 61)	*p*	Cohen’s *d*
Delta	4.75 ± 1.03	4.72 ± 1.02	0.118	0.906	0.03
Theta	4.27 ± 0.89	4.64 ± 0.97	−1.582	0.119	0.41
Alpha1	2.86 ± 0.67	2.95 ± 0.92	−0.464	0.644	0.12
Alpha2	2.72 ± 0.57	2.73 ± 0.63	−0.044	0.965	0.01
Beta1	3.92 ± 0.71	3.91 ± 0.70	0.069	0.945	0.02
Beta2	4.63 ± 0.92	5.21 ± 0.93	−2.452	**0.017**	0.63
Gamma	5.38 ± 1.34	6.21 ± 1.24	−2.546	**0.013**	0.65

Bold values indicate statistical significance (*p* < 0.05).

**Table 4 ijerph-19-01778-t004:** SVM classification while discriminating ScD and HCs groups.

EEG Features	Linear Kernel Function		Gaussian Kernel Function	
	Accuracy	AUC	Accuracy	AUC
Microstate	62%	67%	63%	69%
Omega complexity	60%	62%	60%	64%
Combine	75%	80%	76%	81%

AUC: area under curve.

## Data Availability

The study data can be accessed from the corresponding author A.C. by request.

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
