# Peer review of "Temporal and Spatial Dynamics of EEG Features in Female College Students with Subclinical Depression"

_ijerph, 2022, doi:10.3390/ijerph19031778_

Round 1

Reviewer 1 Report

In the article "Temporal and spatial dynamics of EEG features in female subclinical depression", the authors use EEG to discern novel biomarkers of subclinical depression in students. Overall, I think that this is an important study that warrants for publication. However, much more work is necessary beforehand. I hope that the authors find my suggestions helpful. 

First and foremost, I think that the article does not match the scope of the special issue very well. Only female students are investigated, which does not allow us to draw conclusions on "Gender Differences in Mood Disorders". 

Before resubmission, I think that the following general points should be addressed: 

  1. Making the article better understandable. The theoretical meaning of the measures needs better and more extensive, easier explanations. Some of these explanations are there already, but then should be moved to an earlier position in the article. In the introduction, readers not familiar with the methodology should be equipped with all the necessary information to understand the rest of the article. Among these, of particular importance is the functional meaning of the measures assessed in this study. 
  2. Better neurological explanations. The measures should be explained in close relation with underlying neural processes so that the interpretation of the measures becomes clearer. 
  3. Deriving hypotheses and research methods from theory and previous literature. It seems that there are no clear hypotheses and that there is a lot of literature on temporal dynamics in resting state EEG that remained unmentioned in the present study. 
  4. Better coupling with clinical symptoms. It would be helpful to understand what the functional interpretation of the assessed EEG measures is. Therefore, correlations between the measures and the questionnaire data would be helpful. Furthermore, the inclusion criteria surprised me: the authors claim to study "subclinical" depressive symptoms, yet in the inclusion criteria, only a minimum score on the depression questionnaires is defined, no maximum score. This way, individuals with manifest depressive episodes might be included in the sample. Consequently, the mean BDI in the ScD group is around 25, corresponding to a moderate depressive symptom severity. 
  5. Extensive language edits. Particularly, some of the expressions used in the article seem vague or uncommon, including the unspecific use of terms such as "processes", "mechanisms", or "features". 

Furthermore, I noticed some specific points: 

  1. I think in Table 1, BDI and SDS values are swapped for HC and ScD groups, because the depression scores are reported to be higher in the HC group. 
  2. In the first sentence of the introduction, depressive symptoms are characterized. First, the wording is uncommon, there are fixed English-language terms used to describe depressive symptoms. Second, the reference is Bellón et al., 2020, which is a Spanish public health report, which does not seem to fit the purpose here very well. Better sources would be the APA (diagnostic statistical manual of mental disorders, version 5) or the WHO (international classification of diseases, version 10). 
  3. Figure 1: "Remove of bad trials" is in the figure before ICA, however in the text, the authors state that artifacts were removed by means of ICA. Was there another method of identifying "bad trials" not described in the text or was the figure not adapted to the current analysis? Also, what "trials" are referred to, since the EEG was recorded in a closed-eyes resting state paradigm without a task? 
  4. In Table 2, the variable "transition probability" is not reported, although it is listed above in the text as another microstate characteristic.
  5. On p. 7, the ANOVA factor "transition pairs" is mentioned, but I did not find an explanation what this factor means. 

Reviewer 2 Report

The authors proposed a spatio-temporal EEG analysis based on microstates and omega complexity for studying female subclinical depression. The results showed that the combination of both features improves the classification performance for predicting subclinical depression, as compared with a single index.                                       

The background and the methods are quite clear, the results and the discussion sections are exhaustive. However, I have some comments and suggestions:

  1. Write the extended form of each abbreviation the first time you introduce it in the text.
  2. Add details about the definition of microstates and their four main indexes.
  3. The maximum number of electrodes for the 10-20 system is 21. Maybe do you refer to 10-10 system? (Line 134)
  4. Section 2.6: describe statistical analysis and machine learning in two different sections. Add a brief introduction about machine learning in general and add details about SVM.
  5. In the Discussion section, group the comments about microstates in a subsection (from line 375).

Round 2

Reviewer 1 Report

The authors have done a very thorough work in revising the manuscript, which to my mind has been significantly improved. Unfortunately, however, despite the extensive English language edits (which I think are a great improvement), some aspects about the text still seem difficult to comprehend. In the attached Word file, please find some comments which I hope will prove useful to convey the article’s message in a clearer way.
